# Structural basis for SARS-CoV-2 neutralizing antibodies with novel binding epitopes

**Dan Fu**[1☯], **Guangshun Zhang**[1,2,3☯], **Yuhui Wang**[1,3☯], **Zheng Zhang**[1☯], **Hengrui Hu**[4,5☯], **Shu Shen**[4,6☯], **Jun Wu**[2☯], **Bo Li**[1,3☯], **Xin Li**[1,3,7☯], **Yaohui Fang**[6], **Jia Liu**[4], **Qiao Wang**[8], **Yunjiao Zhou**[8], **Wei Wang**[7], **Yufeng Li**[4], **Zhonghua Lu**[2], **Xiaoxiao Wang**[2], **Cui Nie**[2], **Yujie Tian**[1,9], **Da Chen**[1,3,9], **Yuan Wang**[1,3,9], **Xingdong Zhou**[1,3,9], **Qisheng Wang**[10], **Feng Yu**[10], **Chen Zhang**[1], **Changjing Deng**[2], **Liang Zhou**[2], **Guangkuo Guan**[2], **Na Shao**[2], **Zhiyong Lou**[11]*, **Fei Deng**[4,6]*, **Hongkai Zhang**[1,7,9]*, **Xinwen Chen**[5,12]*, **Manli Wang**[4]*, **Louis Liu**[2]*, **Zihe Rao**[1,3,7]*, **Yu Guo**[1,9,12]*

**1** State Key Laboratory of Medicinal Chemical Biology and College of Pharmacy, Nankai University, Tianjin, China, **2** Harbour Biomed (Suzhou) Co. Ltd., Suzhou Industrial Park, Suzhou, China, **3** College of Life Science, Nankai University, Tianjin, China, **4** Center for Biosafety Mega-Science, Wuhan Institute of Virology, Chinese Academy of Sciences, Wuhan, Hubei, China, **5** Guangzhou Institutes of Biomedicine and Health, Chinese Academy of Sciences, Guangzhou, China, **6** State Key Laboratory of Virology and National Virus Resource Center, Wuhan Institute of Virology, Chinese Academy of Sciences, Wuhan, Hubei, China, **7** Shanghai Institute for Advanced Immunochemical Studies, ShanghaiTech University, Shanghai, China, **8** Key Laboratory of Medical Molecular Virology (MOE/NHC/CAMS), School of Basic Medical Sciences, Shanghai Medical College, Fudan University, Shanghai, China, **9** Frontiers Science Center for Cell Responses, Nankai University, Tianjin, China, **10** Shanghai Synchrotron Radiation Facility, Shanghai Advanced Research Institute, Chinese Academy of Sciences, Shanghai, China, **11** MOE Key Laboratory of Protein Science & Collaborative Innovation Center of Biotherapy, School of Medicine, Tsinghua University, Beijing, China, **12** Guangzhou Laboratory, Guangzhou International Bio-Island, Guangzhou, Guangdong, China

☯ These authors contributed equally to this work.
* louzy@mail.tsinghua.edu.cn (ZL); df@wh.iov.cn (FD); hongkai@nankai.edu.cn (HZ); chen_xinwen@gibh.ac.cn (XC); wangml@wh.iov.cn (MW); Louis.Liu@harbourbiomed.com (LL); raozh@mail.tsinghua.edu.cn (ZR); guoyu@nankai.edu.cn (YG)

**Data Availability Statement:** Protein coordinate and structure factors have been deposited in the RCSB Protein Data Bank under ID 7DEO (PR1077), 7DET (PR961) and 7DEU (PR953).

## Abstract

The ongoing Coronavirus Disease 2019 (COVID-19) pandemic caused by Severe Acute Respiratory Syndrome Coronavirus 2 (SARS-CoV-2) threatens global public health and economy unprecedentedly, requiring accelerating development of prophylactic and therapeutic interventions. Molecular understanding of neutralizing antibodies (NAbs) would greatly help advance the development of monoclonal antibody (mAb) therapy, as well as the design of next generation recombinant vaccines. Here, we applied H2L2 transgenic mice encoding the human immunoglobulin variable regions, together with a state-of-the-art antibody discovery platform to immunize and isolate NAbs. From a large panel of isolated antibodies, 25 antibodies showed potent neutralizing activities at sub-nanomolar levels by engaging the spike receptor-binding domain (RBD). Importantly, one human NAb, termed PR1077, from the H2L2 platform and 2 humanized NAb, including PR953 and PR961, were further characterized and subjected for subsequent structural analysis. High-resolution X-ray crystallography structures unveiled novel epitopes on the receptor-binding motif (RBM) for PR1077 and PR953, which directly compete with human angiotensin-converting enzyme 2 (hACE2) for binding, and a novel non-blocking epitope on the neighboring site near RBM for PR961. Moreover, we further tested the antiviral efficiency of PR1077 in the Ad5-hACE2

**Funding:** Y.G and H.Z are employees of Nankai University, and receive salaries from Nankai University. This study was funded by the National Program on Key Research Project of China 2018YFE0200402 (Y.G), 2017YFC840300 (Z.R), 2018YFA0507203 (Z.R), and 2017YFA0504801 (H. Z) http://www.most.gov.cn/eng/eng/; the National Natural Science Foundation of China (NSFC) 31670731 (Y.G) and 31870733 (Y.G) http://www. nsfc.gov.cn/english/site_1/index.html; Projects of International Cooperation and Exchanges NSFC grant no. 81520108019 (Z.R) http://www.nsfc.gov. cn/english/site_1/index.html; the Key Project of Tianjin Municipal Natural Science Foundation of China 20JCYBJC01340 (Y.G) http://kxjs.tj.gov.cn/; the Science and Technology Innovation Achievements and Team Building Foundation of Nankai University grant no. ZB19500403, ZB19100123 and 63201101 (Z.R) www.nankai. edu.cn and the Emergency Key Program of Guangzhou Laboratory grant no. EKPGL2021008 (Y.G.) http://www.bio-island.com/. The funders had no role in study design, data collection and analysis, decision to publish, or preparation of the manuscript.

**Competing interests:** I have read the journal's policy and the authors of this manuscript have the following competing interests: Y.G., Z.R., L.L., J. W., Z.L., D.F., X.W. and C.N. are inventors in a pending patent application filed on the reported antibodies. L.L., J.W., Z.L., X.W., C.N., C.D., L.Z., G. G. and N.S. are employees of Harbour Biomed (Suzhou) Co. Ltd. Other authors declare no competing interests. All reagents and information presented in this study are available from corresponding authors upon reasonable request.

**Abbreviations:** ABSL-3, Animal Biosafety Level 3; ADE, antibody-dependent enhancement; AHC, anti-human Fc; BLI, bio-layer interferometry; BSA, buried surface area; BSL-3, Biosafety Level 3; cDNA, complementary DNA; CDR, complementarity-determining region; CFA, Complete Freund's Adjuvant; COVID-19, Coronavirus Disease 2019; DAD, diffuse alveolar damage; d.p.i., days post-infection; ELISA, enzyme-linked immunosorbent assay; FFPE, formalin fixed, paraffin embedded; hACE2, human angiotensin-converting enzyme 2; HC, heavy chain; HCDR, heavy chain complementarity-determining region; HE, hematoxylin–eosin; h.p.i., hours post-infection; IACUC, Institutional Animal Care and Use Committee; IMGT, ImMunoGeneTics; LCDR, light chain complementarity-determining region; mAb, monoclonal antibody; NAb, neutralizing antibody; NTD, N terminal domain; qRT-PCR, quantitative real-time PCR; RBD, receptor-binding domain;

transduction mouse model of COVID-19. A single injection provided potent protection against SARS-CoV-2 infection in either prophylactic or treatment groups. Taken together, these results shed light on the development of mAb-related therapeutic interventions for COVID-19.

## Introduction

As of March 22, 2021, the Coronavirus Disease 2019 (COVID-19) has already infected over 124 million individuals and resulted in over 2.7 million deaths worldwide, requiring urgent prophylactic and therapeutic interventions. The etiological agent of COVID-19 is Severe Acute Respiratory Syndrome Coronavirus 2 (SARS-CoV-2) [1]. This emerging coronavirus shares high primary sequence identity with other lineage B coronaviruses and has been categorized into the *Betacoronavirus* genus. SARS-CoV-2 is an enveloped virus with corona-like spike protein protruding from the viral particles. Like other type I transmembrane glycoproteins, the spike (S) glycoprotein forms homotrimers and is primed via cleavage into S1 and S2 subunits. S1 comprises the receptor-binding domain (RBD) and the N terminal domain (NTD), which mediate the recognition of and binding to the human receptors, including human angiotensin-converting enzyme 2 (hACE2) [2] and other potential co-receptors [3]. Tomography image indicated that more than half of the spike trimers adopt the "all close" state on SARS-CoV-2 particles [4]; interaction with cellular receptor hACE2 requires and locks the RBD into the "up" conformation, inducing virus–host cell membrane fusion engaged by the S2 subunit. The SARS-CoV-2 spike protein, especially the RBD region, serves as the primary target for interfering with the virus entry process [5].

Along with the development of vaccines and small molecule drugs, neutralizing antibodies (NAbs) against SARS-CoV-2 have also been proven to be an effective countermeasure for both prophylactic and therapeutic interventions, especially for seniors or individuals at high risk of viral infection. Many efforts have been devoted to the isolation and validation of SARS-CoV-2 NAbs, mainly from COVID-19 convalescent patients [6–17]. Besides helping in the development of monoclonal antibody (mAb) therapy, molecular understanding of NAbs would also greatly accelerate the design of next generation recombinant vaccines [17,18]. However, although numerous NAbs have been reported, most fall into several dominant germlines and recognize similar epitopes. For instance, hACE2-blocking NAbs composed of heavy chains (HCs) encoded by VH3-53 [1–4] and VH3-66 [1,2,5,6] exhibit a common RBD binding pattern featured by a shorter CDRH3 than the average length in human antibodies. The limited knowledge concerning more critical epitopes and immunogenic features of SARS-CoV-2 hampers the development of effective antibody-related therapeutics against COVID-19.

Here, we utilized the H2L2 transgenic mice platform, which represents an alternative source of human antibodies, along with a state-of-art antibody screening platform, to identify NAbs for SARS-CoV-2. Among the 127 isolated antibodies, 25 RBD-specific antibodies showed potent neutralizing activities with sub-nanomolar $IC_{50}$. The atomic-level structural information provides deep insights into novel epitopes on the RBD, with PR1077 and PR953 binding sites overlapping with the binding site of hACE2, and the hACE2 non-blocker antibody PR961 recognizing the neighboring site near RBM. In addition, we further assessed PR1077 for its potential in vivo efficacy in an animal model, which validated its potent protection against SARS-CoV-2 infection, both prophylactically and therapeutically.

RBM, receptor-binding motif; RLU, relative luminescence unit; RT, reverse transcriptase; SARS-CoV-2, Severe Acute Respiratory Syndrome Coronavirus 2; scFv, single-chain variable fragment; SSRF, Shanghai Synchrotron Radiation Facility; QC, quality control.

# Results

## Mouse immunization and antibody isolation

The Harbour H2L2 transgenic mouse carries fully human V-region heavy and light chains and rodent constant regions to allow endogenous affinity maturation and immune effector function, which features an immune response comparable to normal mice and offers diverse human V-gene usage [7]. For the rapid generation of high-quality SARS-CoV-2 NAbs, both Harbour H2L2 transgenic mice and BALB/c mice were immunized with the RBD domain of SARS-COV-2's spike protein [8]. Sera were collected before and after each vaccination, and serum binding to RBD was determined by enzyme-linked immunosorbent assay (ELISA). After 3 to 4 boosts, the spleen and bone marrow cells were collected from immunized mice with high serum titers, and plasma B cells were isolated.

Next, plasma B cells were loaded onto the Berkeley Lights Beacon Optofluidic system mounted with the OptoSelect chip, and assays were run to select single plasma B cells secreting SARS-CoV-2 RBD protein-specific antibodies [9,10] (Fig 1A, S1 Fig, S1 Table). Antigen-reactive B cells were exported to 96-wells preloaded with lysis buffer, followed by single cell sequencing designed to extract sequences of the VH (variable region, HC) and VL (variable region, light chain), which were paired to obtain the antibody sequence.

From 9 single B cell cloning experiments, we identified 105 antibody sequences from Harbour H2L2 transgenic mice and 191 from BALB/c mice in total. We synthesized the genes of these antibodies and transiently expressed them in HEK293T cells in 24-well plates. The supernatant of transient expression cells was collected and tested for its binding to SARS-CoV-2 S1, as well as the effect on SARS-CoV-2 RBD binding to the hACE2 protein (Fig 1B and 1C). Primarily based on binding activity in ELISA ($OD_{450} > 0.5$), 56 mAbs from Harbour H2L2 transgenic mice and 71 mAbs from BALB/c mice were selected for production in HEK293 system, followed by purification, sequence analysis, and further evaluation (Fig 1D).

## Selection and characterization of the purified antibodies

The binding activity to SARS-CoV-2 S1 protein and blocking activity against SARS-CoV-2 RBD binding to hACE2 of the purified antibodies were tested. Meanwhile, cross-reactivity to SARS S1 and MERS S1 proteins were also assessed by ELISA (Figs 1B, 1C, and 2A–2D). The $EC_{50}$ of antibody binding to SARS-CoV-2 S1 was as low as approximately 1.5 ng/mL. Among the 56 mAbs from Harbour H2L2 transgenic mice and 71 mAbs from BALB/c mice that bound to SARS-CoV-2 S1 protein, 13 mAbs from H2L2 mice and 23 mAbs from BALB/c mice could block binding of soluble SARS-CoV-2-RBD to hACE2 protein, with $IC_{50}$ values ranging from 74 ng/ml to 338 ng/mL for mAbs from H2L2 mice and from 5.5 ng/mL to 139.8 ng/mL for mAbs from BALB/c mice (Fig 2D, S1 Table). The binding kinetics of these mAbs were further assessed by bio-layer interferometry (BLI) assays, and equilibrium constant ($K_D$) values were calculated. $K_D$ values for SARS-CoV-2 S1 were mostly at the nM or sub-nM level, while those for SARS-CoV-2-RBD were mostly at sub-nM or better (S1 Table).

The neutralizing activities of all 127 mAbs were further investigated by assessing pseudoviruses and authentic viruses in vitro, respectively. Totally, 25 NAbs were identified with $IC_{50}$ values from 6.8 ng/mL to 3,071 ng/mL in pseudovirus based neutralization assays and from 14.4 ng/mL to approximately 1,753 ng/mL in live virus–based neutralization assays (Fig 1B and 1C, S1 Table). PR1077 from H2L2 transgenic mice and PR953 from BALB/C had the most potent neutralizing activities against live SARS-CoV-2 infection of Vero-E6 cells, with $IC_{50}$ values of 27.9 and 14.4 ng/mL, respectively. Among the antibodies without hACE2-blocking capacity, PR961 was the most potent neutralizing with its $IC_{50}$ of 570 ng/mL (S2E and S2F and S3 Figs).

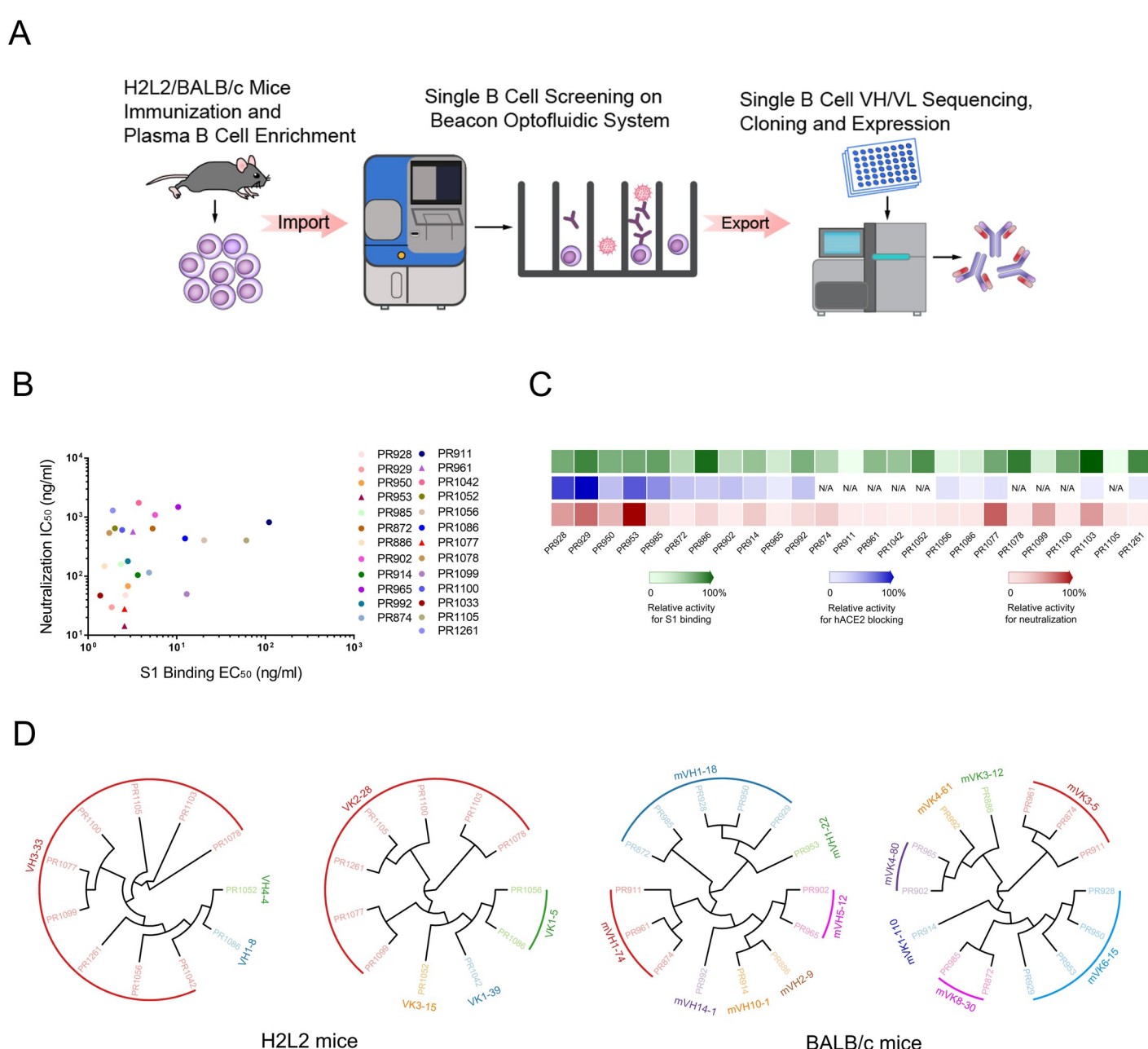

**Fig 1. Analysis of plasma responses to SARS-CoV-2 proteins and antibody identification from H2L2 transgenic and BALB/c mice by single cell sequencing. (A)** Schematic diagram of antibody identification from convalescent patients by single cell sequencing. Both Harbour H2L2 transgenic mice and BALB/c mice were immunized with SARS-CoV-2 RBD protein. SARS-CoV-2 RBD protein binding B cells were isolated from enriched mouse spleen and bone marrow cells with beads conjugated with biotinylated SARS-CoV-2 RBD protein. Paired VH and VL of each cell were recovered by single B cell sequencing. The sequences were used for subsequent antibody construction and expression. **(B)** Antibodies neutralization $IC_{50}$ ($y$ axis) are plotted against SARS-CoV-2 S1 protein binding $EC_{50}$ ($x$ axis), PR1077, PR953, and PR961 are shown as triangle, and other antibodies are shown as dot. All the data of this figure can be found in the S1 Data file. **(C)** Heat map shows the relative levels of SARS-CoV-2 S1 binding, hACE2 blocking, and live virus neutralization. The depth of color is proportional to the effect of the antibody (green, S1 binding activity; blue, hACE2 blocking activity; red, neutralizing activity). All the data of this figure can be found in the S2 Data file. **(D)** Phylogenetic analysis of the relationship between antibody variable gene segments and germline. The relationships between the heavy and light chain variable regions and the germlines from H2L2 transgenic (left) and BALB/c (right) mouse antibodies are shown. hACE2, human angiotensin-converting enzyme 2; RBD, receptor-binding domain; SARS-CoV-2, Severe Acute Respiratory Syndrome Coronavirus 2.

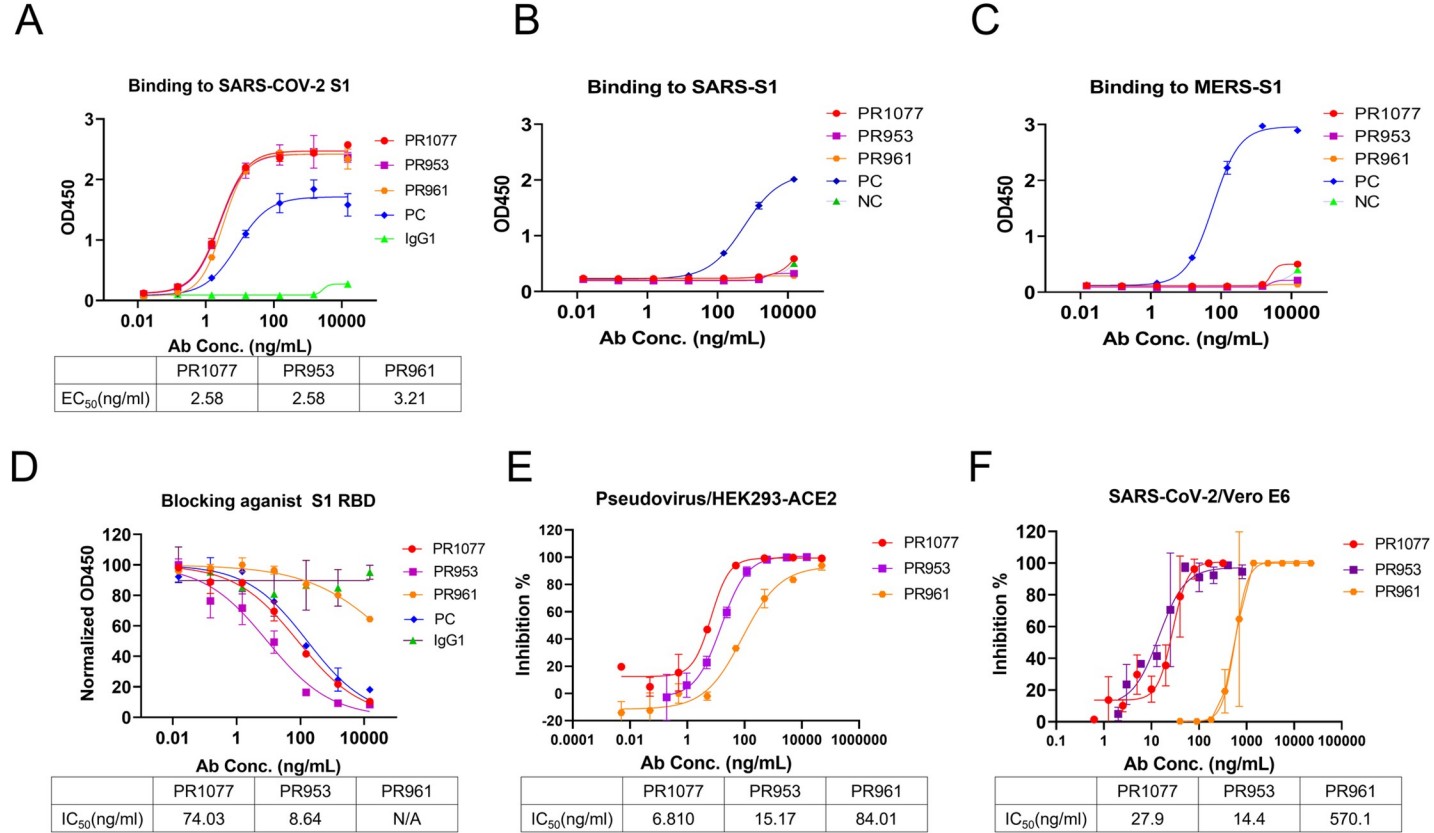

**Fig 2. Characterization of the selected NAbs PR1077, PR953, and PR961. (A)** ELISA binding curve of SARS-CoV-2 S1 protein. A commercially available antibody (8A5, Novoprotein, Shanghai) was applied as a PC. Normal human IgG1 was used as a negative control. All the data of this figure can be found in the S3 Data file. ELISA binding curves of SARS S1 **(B)** and MERS S1 **(C)** proteins. Internally generated antibodies were used as PCs. All the data of these figures can be found in the S4 Data and S5 Data files. **(D)** ELISA-based RBA of PR1077, PR953, and PR961 against RBD binding to hACE2. All the data of this figure can be found in the S6 Data file. **(E)** PR1077, PR953, and PR961 could effectively neutralize SARS-CoV-2 pseudovirus in vitro. SARS-CoV-2 pseudovirus was incubated with serially diluted antibodies. The mixture was added to HEK293-hACE2 cells for 48 hours. The neutralization potency of each antibody was evaluated in a luciferase assay system. All the data of this figure can be found in the S7 Data file. **(F)** PR1077, PR953, and PR961 showed strong neutralizing potency against live SARS-CoV-2 virus in vitro. The mixture of live SARS-CoV-2 virus and serially diluted PR1077 were added to Vero E6 cells. After 1 hour, cells were washed and further incubated for 48 hours before detection of infected cells by an immunofluorescence assay. All the data of this figure can be found in the S8 Data file. ELISA, enzyme-linked immunosorbent assay; NAb, neutralizing antibody; PC, positive control; RBA, receptor blocking assay; SARS-CoV-2, Severe Acute Respiratory Syndrome Coronavirus 2.

According to ImMunoGeneTics (IMGT) database-based analysis, the germlines of these 25 NAbs were defined. The VH domains of 11 NAbs from H2L2 mice were encoded by 3 different germlines, while their VL domains belonged to 4 different germlines. Most of the antibodies from H2L2 mice, including the most potent antibody PR1077, were encoded by the same germline origin of VH3-33 and VL2-28. Given the high primary sequence similarities between VH3-33 and VH3-30 germlines, this result is consistent with the previous finding that VH3-33 is one of the most frequently used germlines for NAbs isolated from patients recovering from SARS-CoV-2 infection [11–13,15,19]. The germlines of BALB/c mouse-derived antibodies showed a higher diversity. The VH and VL regions of 14 mouse antibodies were assigned to 7 IGHV and 8 IGLV germlines. Notably, the most potent mouse antibody PR953 was assigned to mVH1-22 and mVK6-15, and the germline encoding the non-blocking NAb PR961 was mVH1-74 and mVK3_5 (Fig 1C and 1D). These antibodies were then classified into 4 groups according to the competitive ELISA results (S4 Fig). Subsequently, based on sequence similarities with related human germlines, PR953 and PR961 were engineered as humanized antibodies.

## PR1077, PR961, and PR953 recognize novel epitopes of SARS-CoV-2 RBD

To gain further insights into the structural basis of the blocking and neutralizing mechanism, we selected 2 potent hACE2-blocking antibodies (PR1077 and PR953), alongside 1 non-blocker PR961, for subsequent structural analysis. The structure of the antibody Fab/Spike-RBD complex was resolved by X-ray crystallography at the resolutions of 2.5, 2.1, and 2.2 Å, respectively (Fig 3A, S2 Table). To the best of our knowledge, all 3 mAbs recognized novel epitopes of SARS-CoV-2 spike RBD. Overall, PR1077 and PR953 interacted with RBD near the hACE2 binding site, and hence blocked the binding of the cellular receptor hACE2 (Fig 3B,

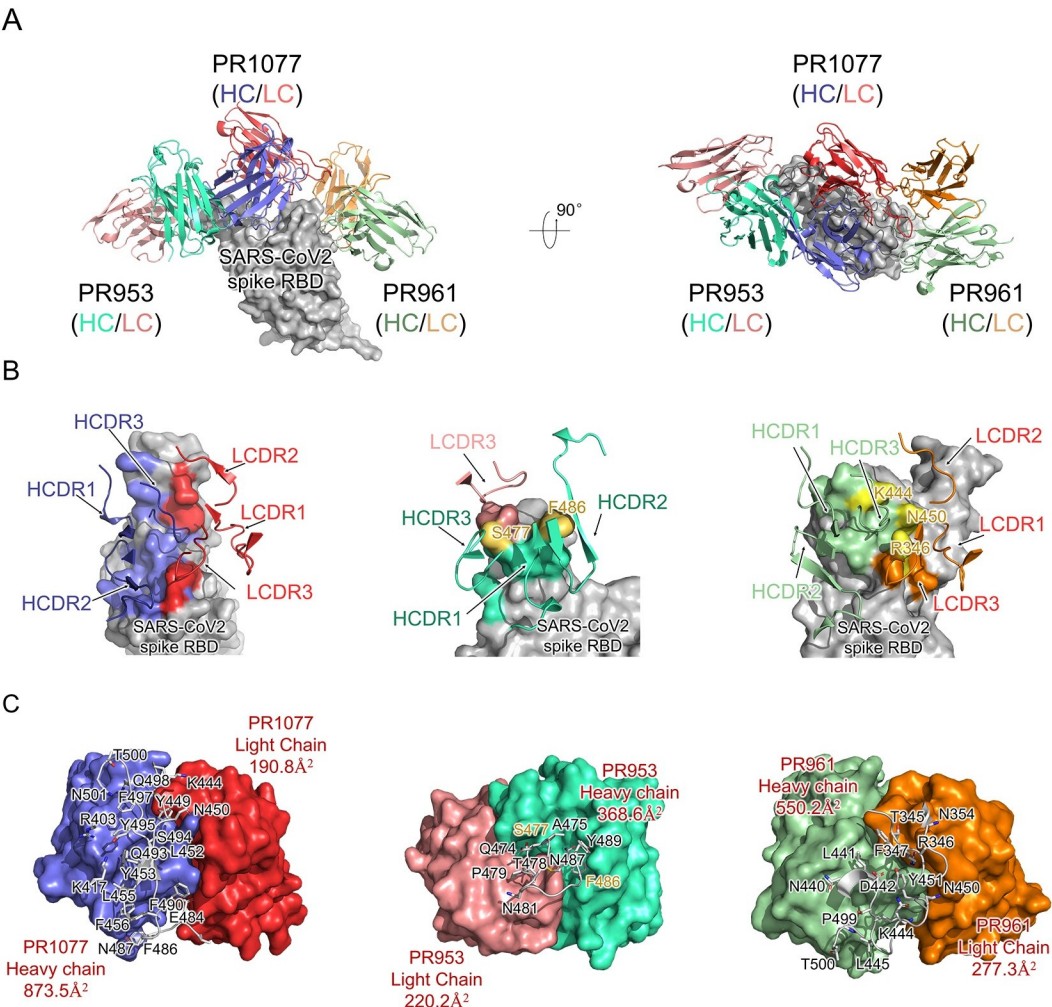

**Fig 3. Structural analysis of the SARS-CoV-2 RBD-NAbs scFv complex. (A)** Overall structures of the PR1077, PR953, and PR961-Fab-RBD complexes. The PR1077 HC (colored slate blue) and light chain (colored red) are shown. PR953 heavy (colored green cyan) and light (colored salmon red) chains and PR961 heavy (colored pale green) and light (colored orange) chains are displayed. The SARS-CoV-2 RBD is colored in gray and displayed in surface representation. **(B)** The epitopes of antibodies are shown in surface representation. The CDR loops of NAbs are colored as above. The S477 and F486 residues in contact with both the heavy and light chains of PR953 are colored in yellow; 3 hydrogen donor residues (R346, K444, and N450) which contact with both the heavy and light chains of PR961 are colored in yellow. **(C)** The binding surface of NAbs with SARS-CoV-2-RBD. The interaction residues of RBD are shown in sticks and labeled accordingly; variable regions' interaction surfaces of NAbs are colored as above. HC, heavy chain; HCDR, heavy chain complementarity-determining region; LCDR, light chain complementarity-determining region; NAb, neutralizing antibody; RBD, receptor-binding domain; SARS-CoV-2, Severe Acute Respiratory Syndrome Coronavirus 2; scFv, single-chain variable fragment.

left and middle panels). In contrast, PR961 recognized RBD a novel epitope that is adjacent to the receptor-binding motif (RBM), making PR961 a non-blocking NAb for SARS-CoV-2 (Fig 3B, right panel).

Specifically, PR1077 is located directly at the RBM region of SARS-CoV-2 Spike RBD; the buried surface areas (BSAs) of the heavy and light chains were 873.5 Å$^2$ and 190.8 Å$^2$, respectively, inducing steric hindrance to the binding of receptor hACE2 (Fig 3C, left panel). All the heavy chain complementarity-determining regions (HCDRs) and light chain complementarity-determining regions (LCDRs) participate in the interaction with SARS-CoV-2 RBD, mainly contributed by polar interactions and hydrogen bonds. F27, Y31, Y32, W52, Y53, F59, P101, L103, and F105 of the HCDRs and Y35, Y37, and Y54 of the LCDRs form strong hydrophobic interactions with the hydrophobic rim constituted by Y449, Y453, L455, F456, F486, Y489, F490, L492, Y495, and Y505. In addition, interactions were also facilitated by the hydrogen bond network constituted by Y31, Y32, W52, Y53, D54, S56, N57, R58, G101, G102, R104, and R106 of the HCDRs and H31, N33, Y35, Y54, Y37, and T99 of the LCDRs. It is worth noting that water molecules also played an important role in LCDR interactions with SARS-CoV-2 RBD (Fig 4A).

Despite the potent binding, blocking, and neutralizing activities of PR953, its BSA (368.6 Å$^2$ for the HC and 220.2 Å$^2$ for the light chain) was much lower than expected. The complementarity-determining regions (CDRs) of PR953 formed a deep concave binding pocket and tightly covered the flexible tip of the RBM region (I472 to F490), partially overlapping with the hACE2 binding site (Fig 3C, middle panel). The interactions of PR953 were mainly stabilized by an extensive hydrogen bond network. T33, H35, N52, N55, D57, T59, D99, Y101, and Y105 of the HCDRs recognized the bulge region constituted by Q474, A475, G476, S477, N487, and Y489. The interactions were further stabilized by electrostatic interactions among Y91, N92, N93, Y94, and W96 of the LCDRs with S477, S478, N481, and F486 (Fig 4B).

In contrast, PR961 recognized SARS-CoV-2 RBD at the opposite side of PR953; this epitope was adjacent to the receptor-binding motif and did not induce direct steric hindrance to the binding of receptor hACE2, in agreement with binding and competition assays. The BSAs of the heavy and light chains were 550.2 Å$^2$ and 277.3 Å$^2$, respectively, mainly contributed by electrostatic forces (Fig 3C, right panel). S31, W33, H52, S54, D55, E57, D100, G101, Y102, and E103 of the HCDRs and S31, Y32, G33, N34, R54, S95, N96, and E97 of the LCDRs formed strong electrostatic interactions with the novel epitope constituted by T345, R346, F347, N440, S443, K444, V445, and N450. It is worth noting that 3 hydrogen donor residues, i.e., R346 and K444, along with N450, protruded at the RBD surface and interacted with both the heavy and light chains of PR961, playing a critical role in the interactions (Fig 4C).

## Characterization of PR1077 as potential therapeutic candidate

Given its unique epitope, high potency of SARS-CoV-2 RBD binding, and neutralizing activity on live viruses, PR1077 was subsequently selected for further investigation as a potential therapeutic candidate against COVID-19.

Recent tomography investigation of SARS-CoV-2 virus particles revealed 54% of the spike adopt "down" conformation [16]. Most of the identified antibodies could bind to SARS-CoV-2 spike trimer when RBDs are in the "open form"; in contrast, few antibodies could interact with spike trimer when all RBDs are in the "closed form" [20], since the antibody binding epitope of on RBM would be partially covered by the neighboring spike protomer for some antibodies. To analyze its potential binding mode on SARS-CoV-2 spike trimer, PR1077 was superimposed to the spike trimer structure. Interestingly, PR1077 epitope was shown to be fully accessible in either the open or "all close state" spike trimer (S5 Fig). More structural

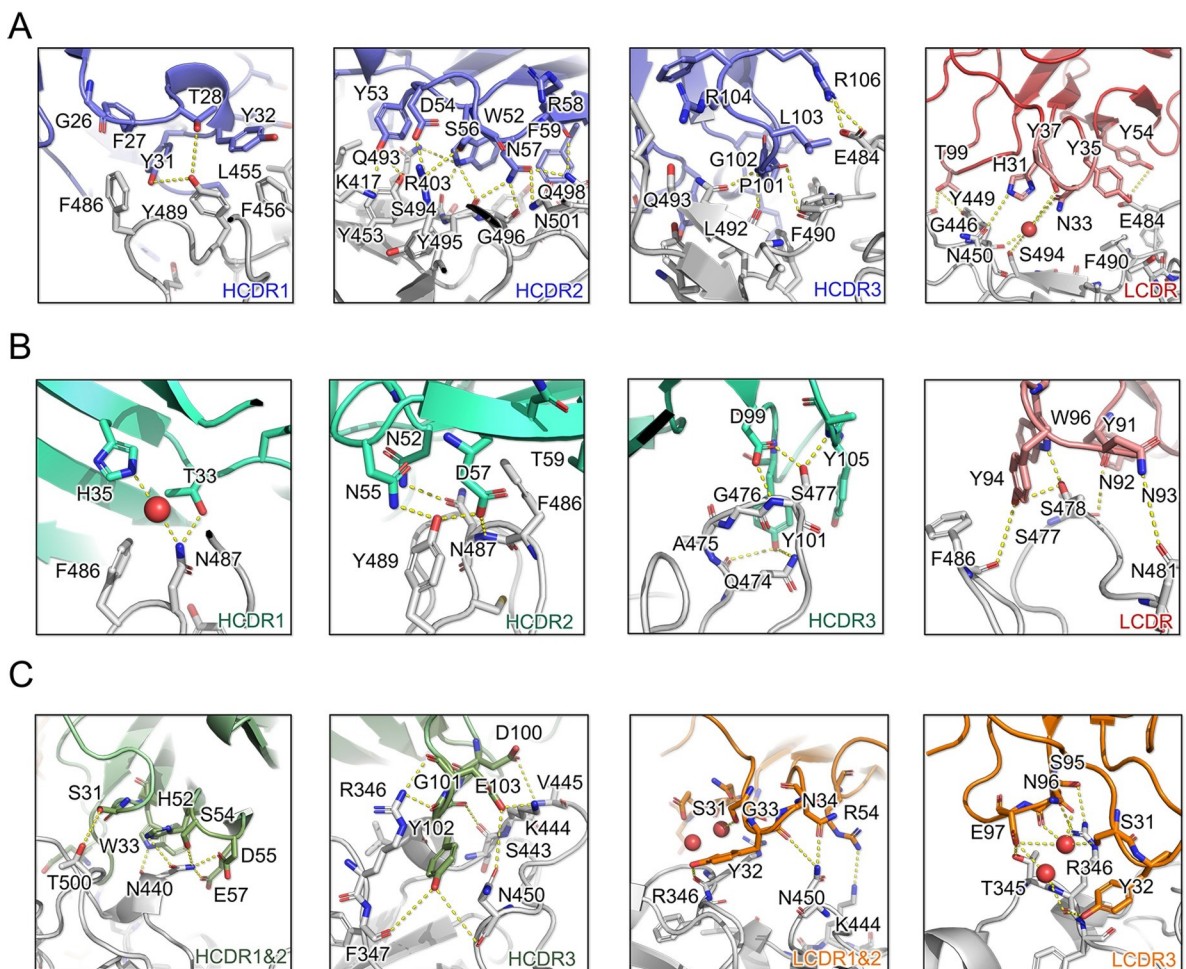

**Fig 4. Interactions between the SARS-CoV-2 RBD and CDR loops. (A–C)** Detailed analysis of the interfaces of SARS-CoV-2 RBD with HCDR and LCDR for (A) PR1077, (B) PR953, and (C) PR961. For clarity, only the key residues are labeled: red, oxygen atoms; blue, nitrogen atoms; yellow dash lines, hydrogen bond interactions. The residues are shown in sticks with identical colors to Fig 3; water molecules are displayed as red spheres. CDR, complementary-determining region; HCDR, heavy chain complementarity-determining region; LCDR, light chain complementary-determining region; RBD, receptor-binding domain; SARS-CoV-2, Severe Acute Respiratory Syndrome Coronavirus 2.

data, such as the CryoEM structure of the spike-PR1077 complex, would further support this hypothesis.

Next, to test the possibility that some of the evolving mutations might confer antibody resistance, we measured the binding activity of PR1077 with RBD mutants and neutralizing activity against SARS-CoV-2 mutants by in vitro pseudovirus assays. All the tested clinical mutations remained sensitive to PR1077, with $EC_{50}$ values from 2.6 ng/mL to 700 ng/mL in ELISA binding assays (S5 Fig), and with similar $IC_{50}$ values ranging from 5.6 ng/mL to 18.6 ng/mL in neutralization assays of pseudoviruses expressing wild-type or mutated spike protein. In agreement with the potent binding affinity and broad coverage of PR1077 epitopes, these results indicated the robustness of PR1077 against viral escape.

Another aspect to consider in the development of vaccine and NAb therapies is the potential risk of antibody-dependent enhancement (ADE) of infection. Although the existence of ADE in COVID-19 patients remains elusive, the ADE effect in vitro has been reported recently, and it is of great importance to circumvent this potential risk [17]. Therefore, we

used Raji cells, which represent a FcγRII-bearing human B lymphoblast cell line, to study the antibody-dependent viral entry as previously reported [18]. The antibody-dependent entry of SARS-CoV-2 pseudovirus in the presence of PR1077 was measured, and no ADE effect was detected at various antibody concentrations (S6 Fig).

### In vivo efficacy evaluation of PR1077

To facilitate its clinical application, the neutralization activity of PR1077 against SARS-CoV-2 infection in vivo was tested in AdV-hACE2-Transduced IFNAR$^{-/-}$ mice, which show high susceptibility to SARS-CoV-2 infection [19,20]. First, the mice were intranasally transduced with $4 \times 10^8$ TCID$_{50}$ of Ad5-hACE2 for hACE2 expression in the lungs. Five days post-transduction, the animals were challenged with $1 \times 10^6$ TCID$_{50}$ of SARS-CoV-2 via the intranasal route and monitored over a 6-day time course (Fig 5A). For prophylactic treatment, mice were intraperitoneally injected with 50 mg/kg of PR1077 or 200 μl PBS at 24 hours before SARS-CoV-2 infection. The mice lost about 15% of the body weight in the first 2 days post-infection (d.p.i.), and prophylaxis with PR1077 led to almost complete recovery of the body weight loss and significantly reduced viral RNA levels in the lungs as determined by quantitative real-time PCR (qRT-PCR) (Fig 5B). For therapeutic treatment, the mice were treated with PBS or different doses of PR1077 (50 mg/kg or 25 mg/kg) at 2 hours post-infection (h.p.i.). As shown in Fig 5C, both PR1077 doses could prevent severe SARS-CoV-2–induced weight loss in infected mice after 3 d.p.i. and reduce viral RNA levels in the lungs at 6 d.p.i. (Fig 5C).

Hematoxylin–eosin (HE) staining showed that the lung samples from control mice displayed severe diffuse alveolar damage (DAD), including thickening of alveolar septa, extensive fibrosis with exudation of fibrin and protein edema in the alveoli, marked epithelial hyperplasia in the bronchi/bronchioles, and extensive immune infiltration in the alveoli, peri-bronchi and peri-vessels at 6 d.p.i. By contrast, in mice pretreated with PR1077, most alveolar septa and cavities were stained normally, and only small amounts of immune cell infiltration around bronchi/bronchioles and blood vessels were observed (Fig 5D). Similar to the prophylactic group, mice treated with 50 mg/kg of PR1077 also displayed mild pathological changes. In mice treated with 25 mg/kg of PR1077, moderate levels of DAD, peri-bronchi and peri-vascular cuffing, and interstitial inflammation were detected compared with the control group (Fig 5E). Therefore, PR1077 could effectively protect mice against SARS-CoV-2 infection either prophylactically or therapeutically.

### Discussion

Since not every infected individual generates high activity NAbs, mAb therapy would not only elicit optimized therapeutic potency, but also constitutes an important prophylactic intervention to safeguard the population at high risk of viral infection as complementary to vaccine approaches. Moreover, the characterization of NAbs, especially molecular understanding of atomic resolution, would provide valuable information for rational vaccine design [21]. Recently, many efforts have been devoted to the structural investigation of NAbs (S7 Fig). IGHV 1–2, 1–46, 3–9, 3–30, and 3–53 are among the most frequently used IGHV genes from convalescent patients [1,2,4,11–13,15,22–24]. Here, we isolated and identified 25 NAbs of sub-nanomolar IC$_{50}$ from H2L2 or traditional mice platforms. Among them, 3 potent NAbs with novel epitopes were well studied by high-resolution structure elucidation, including 2 potent hACE2-blocking antibodies (PR1077 and PR953) and a non-blocking antibody (PR961). The variable region of PR953 forms a deep concave binding pocket and facilitates tight binding with the flexible tip of the RBM region. Despite its relatively small epitope, PR953 presented potent binding affinity and neutralizing activity by overlapping with the hACE2 binding site.

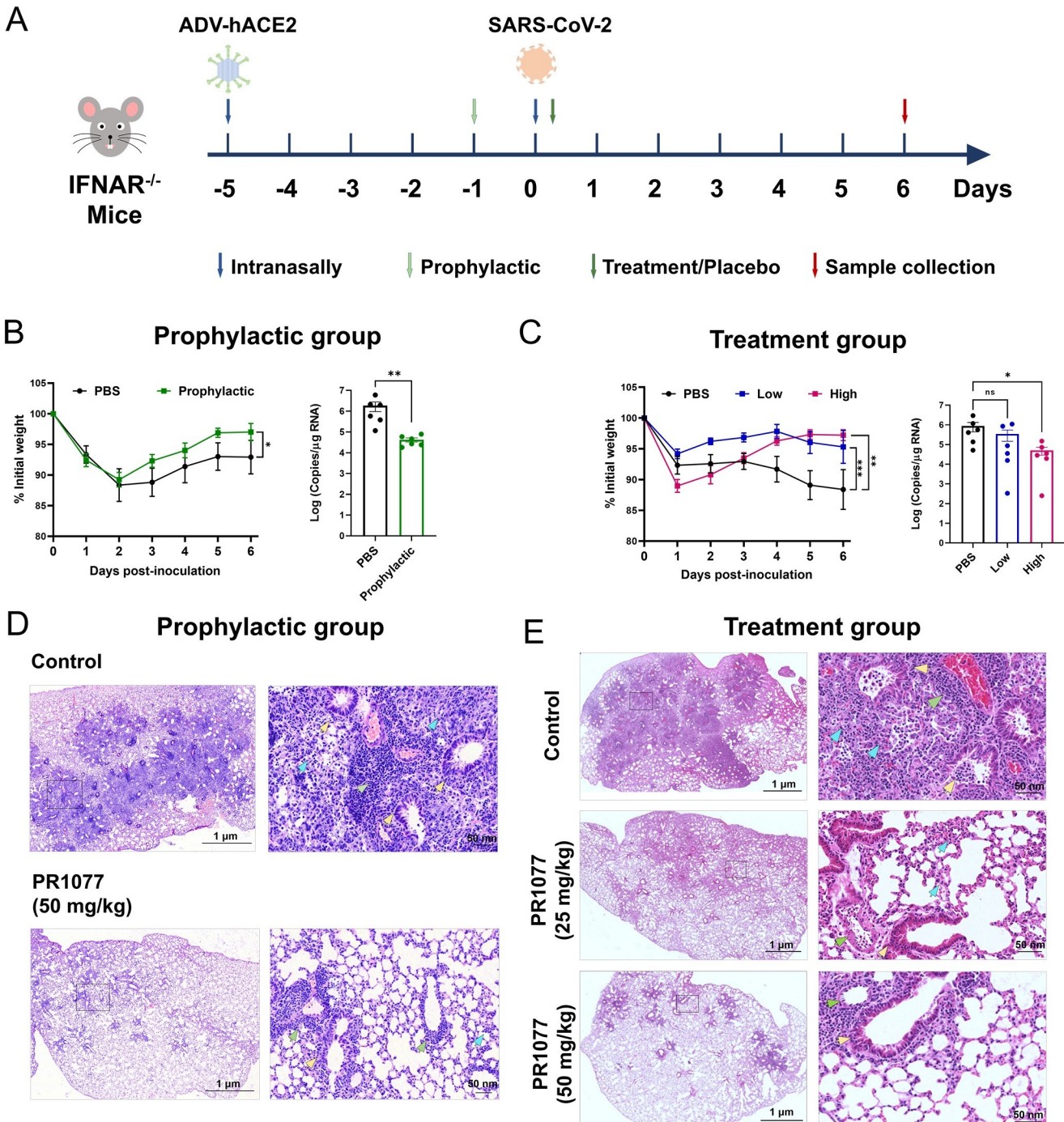

**Fig 5. Therapeutic efficacy of PR1077 in SARS-CoV-2–infected AdV-hACE2-transduced mice. (A)** Experimental design for PR1077 neutralization activity testing in AdV-hACE2-transduced IFNAR$^{-/-}$ mice. **(B, C)** Body weight changes and viral copies in the lung of infected mice were monitored. (B) For prophylactic efficacy testing, mice were intraperitoneally injected with 50 mg/kg of PR1077 or 200 μl PBS (*n* = 6 per group) at 24 hours before SARS-CoV-2 infection. All the data of this figure can be found in the S9 Data file. (C) For therapeutic efficacy testing, the animals were treated with PBS, 50 mg/kg, or 25 mg/kg of PR1077 at 2 h p.i. (*n* = 6 per group). Weight changes were monitored daily, and viral copies in the lungs were measured at 6 d p.i. by qRT-PCR. All the data of this figure can be found in the S10 Data file. **(D, E)** Histopathological analyses of PR1077 treated or untreated mice infected with SARS-CoV-2. Representative images of lung sections stained by HE in the PR1077 prophylactic (D) and treatment (E) groups at 6 d p.i. a. Data are mean ± SEM. Blue, yellow, and green arrows indicate the pathological changes in the alveoli, bronchi/bronchioles, and blood vessels, respectively. The images on the right (bars = 1 μm) are enlarged regions in the dashed boxes of the left images (bars = 50 nm). d.p.i., days post-infection; hACE2, human angiotensin-converting enzyme 2; HE, hematoxylin–eosin; h.p.i., hours post-infection; qRT-PCR, quantitative real-time PCR; SARS-CoV-2, Severe Acute Respiratory Syndrome Coronavirus 2.

In contrast, PR961 recognizes another novel epitope, which is located at the opposite side of the PR953 binding site. In consistent with this finding, PR961 do not directly compete with hACE2 binding based on the binding and blocking assay result. The neutralizing effect of PR961 may achieved by S1 shielding and conformational locking of spike trimer. Considering the emergence of new SARS-CoV-2 variants, more investigations on mAb cocktail therapy, which could present synergistic activity and deploy multiple functions to confer broad neutralization, are urgently needed. PR961 may serve as a hit for the optimization and development of non-hACE2 competitive NAb.

An interesting feature of PR1077 is the relatively short HCDR3 sequence. The Cα of P100, P101, and G102 of the HCDR3 is only about 4 Å away from the corresponding RBD residues, indicating that longer HCDR3 sequence, or the side chains of other amino acids, would clash with the RBD if they were present at this location. Sequence analysis further revealed that a high rate of somatic hypermutations is not required for IGHV3-33–derived NAbs (S8 Fig). Considering that IGHV3-30 and IGHV3-33 share high sequence similarity, and the complex structures of NAbs derived from IGHV3-33 with SARS-CoV-2 RBD not available, we compared the primary sequence and three-dimensional structure of PR1077 with those of 3 other NAbs from IGHV3-30 [12,14,25]. To our surprise, despite the high similarity of their HCDRs, alignment of the three-dimensional complex structures revealed significantly different binding modes with each antibody binding SARS-CoV-2 RBD at a distinct epitope. Interestingly, C002 and PR1077 engage SARS-CoV-2 RBD in similar regions, but adopt absolute opposite HCDR and LCDR orientations (S9 Fig). Overall, the molecular understanding of the antibodies presented here not only provide promising guidance for pairing antibody therapy, but also facilitate the rational design of next generation recombinant vaccines.

Notably, PR1077 also displayed broad spectrum of binding affinity against all spike protein mutants tested, as well as excellent neutralizing activity in pseudovirus assay, including the currently prevalent variants D614G, V367F, V483A, and G476S. We further demonstrated that PR1077 displayed no ADE effect and effectively protected mice against SARS-CoV-2 infection, either prophylactically or therapeutically. These results proved that PR1077 may provide durable protection for the elderly or individuals at high risk of COVID-19 infection.

Overall, the characterizations of these NAbs revealed novel and complementary epitopes on the SARS-CoV-2 RBD, providing potential pairing strategy for development of mAb cocktails against viral resistance that might arise under selective pressure by single antibody therapy. We anticipate that the comprehensive structural and functional characterization of these antibodies would provide a promising starting point for the development of more COVID-19 countermeasures in humans.

## Materials and methods

### Ethics statement

Mice immunization experiments were approved by the Institutional Animal Care and Use Committee (IACUC) of Harbour Biomed (Ethics number: AN-2020-E-0708) and conducted within the animal facility in the HUST-Suzhou Institute for Brainsmatics. The in vivo efficacy experiments were approved by the IACUC of Wuhan Institute of Virology, Chinese Academy of Sciences (Ethics number: WIVA01202001) and conducted within the Animal Biosafety Level 3 (ABSL-3) facility in the National Biosafety Laboratory (Wuhan), Chinese Academy of Sciences.

### SARS-CoV-2 spike RBD protein expression and purification

The codon-optimized wild-type cDNA of SARS-CoV-2 RBD (residues 333 to 530) was synthesized. The SARS-CoV-2 RBD with a carboxyl-terminal 8×His tag for purification was cloned

into pAcgp67 vector with *BamH*I and *Not*I restriction sites using the cloning primers. The sequences of the primers were 5′-TCTCCTACATCTACGCCGACGGATCCACCAACCTCTG CCCTTTCGGT-3′ (forward) and 5′-TGGTGATGGTGGTGATGATGTGCGGCCGCACTCTT CTTTGGCCCGCATA-3′ (reverse). The accuracy of the inserts was verified by sequencing. The SARS-CoV-2 RBD was expressed using the Bac-to-Bac baculovirus system. The construct was transformed into bacterial DH5α component cells, and the extracted bacmid was then transfected into Sf9 cells using Cellfectin II Reagent (Invitrogen, USA). The low-titer viruses were harvested and then amplified to generate high-titer virus stock. The viruses and Endo H, Kifunensine, were coinfected Hi5 cells at a density of $2.0 \times 10^6$ cells/ml. The supernatant of cell culture containing the secreted removal of glycosylated RBD was harvested 72 hours after infection, and the recombinant RBD proteins were purified preliminarily on a Ni-NTA affinity column (GE Healthcare, USA). The affinity column was washed with 150 mL of wash buffer (25 mM Tris, 200 mM NaCl, 50 mM imidazole, pH 7.0), and the target protein was eluted with elution buffer containing 25 mM Tris, 200 mM NaCl, 500 mM imidazole, pH 7.0. The protein was further purified on a Superdex 75 16/60 size-exclusion column (GE Healthcare) equilibrated with 25 mM Tris, 200 mM NaCl, pH 7.0. SDS-PAGE analysis revealed over 96% purity of the final purified recombinant protein. Fractions from the single major peak were pooled and concentrated to 30.8 mg/mL.

**Harbour H2L2 transgenic mice and BALB/c mice immunization.** SARS-CoV-2 spike RBD protein was used as immunogen for mice immunization. Quality control (QC) includes concentration, purity, molecular weight measurement, and biological activity test. The SARS-CoV-2 spike RBD protein immunization used 6- to 8-week-old Harbour H2L2 transgenic mice and BALB/c mice (raised by Vital River, China). All mice were fed under SPF condition. The prime immunization was done by injecting 50-μg freshly prepared Complete Freund's Adjuvant (CFA) emulsified SARS-CoV-2 spike RBD protein intraperitoneally. Subsequent boosts were done by injecting 25-μg freshly prepared Ribi adjuvant emulsified SARS-CoV-2 spike RBD protein intraperitoneally. The interval between each immunization is 2 weeks.

**Single B cell screening on Beacon Optofluidic system.** The Beacon Optofluidic system (Berkeley Lights, Emeryville, California, United States of America) is an automated biosystem that could screen thousands of single plasma B cells for antigen-specific antibody-secreting B cells. Briefly, after approximately 3 to 4 boost, spleen and bone marrow cells were collected from immunized mice which showed good immune response based on TB test. Subsequent plasm B cell enrichment was done by using Plasma Cell Isolation Kit II (human, Miltenyi Biotec, USA) and EasySep Human CD138 Positive Selection Kit II (STEMCELL Technologies, USA). In this study, OptoSelect 14K chip was used following manufacture instruction (Berkeley Lights). Enriched plasma B cells were imported into the system and loaded into the nanopen on the chip. SARS-CoV-2 spike RBD conjugated beads mixed with anti-rat IgG (Fc) AF488 secondary antibody (Jackson ImmunoResearch Laboratories, USA) were imported in channel for antigen binding assay with an image capture iterations every 5 minutes for a total of 10 iterations, and antigen-specific antibody-secreting plasma B cells were identified. These antigen-specific antibody-secreting plasma B cells were then exported to 96-well forensic grade PCR plate (Eppendorf, Germany) containing cell lysis buffer (5 μL of TCL buffer (Qiagen, United Kingdom)) supplemented with 5 mM dithiothreitol (Thermo Fisher Scientific, USA), and 15 μL of mineral oil (Sigma, United Kingdom) was finally added to the top of each well. Samples were stored at −80°C until the next step [10].

**Single B cell sequencing to determine heavy chain and light chain complementarity-determining region sequences of H2L2 antibody and CDR domain analysis.** Cell lysate from single B cell screening was undergone RNA purification and reverse transcriptase (RT) PCR using a modified protocol provided by Berkeley Lights. RNA was purified using Agencourt RNAClean XP kit and eluted directly into a RT reaction with Maxima RNaseH minus

RT. A dT primer with adaptor (P1) and a 5′ template switching primer (P2) were used in the RT reaction. The RT reaction product was added to a PCR mix using KAPA HiFi HotStart Readymix with primer P3 in a reaction, amplifying the complementary DNA (cDNA). The amplified cDNA was purified using Agencourt AMPure XP Beads, and then was used to amplify the gamma and kappa chains in separate reactions with primers in the antibody constant regions (CH Primer and CL Primer) and P4 to obtain the specific product for sequencing. The specific PCR products were sequenced at Genewiz with the corresponding antibody constant region primers. Sequence analysis was performed using in-house software and aligned to the international IMGT information system (http://www.imgt.org) database for germline determination.

**Antibody production.** Sequences from the unique variable regions were synthesized with adaptors for cloning at GenScript (China). Cloning was performed using in-house vectors for expression. Plasmids were transfected into human embryonic kidney 293T cells (ATCC) using PEI (Sigma, #24885) in a 24-well plate and incubated at 37˚C, 5% $CO_2$ for 3 days before collecting supernatant for subsequent test. Positive clones in supernatant test were transfected into HEK 293F (Gibco, USA, R79007) or ExpiCHO cells (Thermo Fisher Scientific, A29127) for small-scale recombinant antibody production. Antibody was purified using Protein A (AmMag Protein A Magnetic Beads, Genscript, L00695) affinity chromotography, and purity was measured by both SDS-PAGE (SurePAGE, Bis-Tris, 10 × 8, 4% to 12%, 12 wells, Genscript, M00653) and SEC-HPLC (GE Healthcare).

## ELISA test for antibody characterization

ELISA was used to test the binding between antibody containing supernatant/purified antibodies and the antigen (recombinant SARS-CoV-2 S1 protein) blocking activity of generated anti-SARS-CoV-2 S1 RBD antibodies against recombinant SARS-CoV-2 S1 RBD protein binding to recombinant hACE2 protein and to evaluate the epitope bins of testing antibodies. Briefly, recombinant SARS-CoV-2 S1 protein (Sino Biological, China, 40591-V08H or Genscript, P9FE001) was coated on a 96-well high binding plate overnight at 4˚C, washed with PBST 3 times, incubated with blocking buffer (2% Bovine Serum Albumin) for 1 hour at 37˚C, washed with PBST 3 times, and incubated with antibody containing supernatant or purified antibodies at desired dilutions or concentrations for 1 hour at 37˚C. A secondary antibody, anti-human Fc (AHC)-HRP (Jackson ImmunoResearch Laboratories), was used for detection of the targeted antibody. TMB and stopping solution (Huzhou InnoReagents, China, TMB-S-003) were used for detection of HRP signal. The HRP signal was read on a SPECTRAMAX plate reader. Clones from supernatants with positive signal were selected for further production. For purified antibodies, EC50 was determined from the ELISA test result.

For competitive ELISA, antibody A was bound to the substrate in a 96-well high binding plate. Antibody B and biotinylated SARS-CoV-2 S1 protein were mixed and incubated for 30 minutes before adding to the plated. After 1-hour incubation, an Alexa 488 streptavidin was used to detect the biotinylated SARS-CoV-2 S1 protein bound to antibody A. The percentage of inhibition was calculated.

For receptor blocking assay, antibodies were initially tested with a BSNAT test kit (Bio-Hermes, China, COV-S41). After that, recombinant hACE2 protein was bound to the substrate in a 96-well high binding plate at 5 μg/mL overnight at 4˚C. A mouse Fc-tagged SARS-CoV-2 RBD protein and a testing antibody were added to the plate simultaneously. After 1 hour incubation, an anti-mouse Fc HRP was used to detect the binding of SARS-CoV-2 RBD protein to the hACE2 protein bound to the substrate.

## Affinity measurement by Octet

Antibodies were diluted to 6 μg/mL and immobilized to AHC biosensor, and recombinant SARS-CoV-2 S1 protein/SARS-CoV-2 S1 RBD protein was diluted with 10 X Kinetics buffer, 3 of the following concentration depending on binding activity (800 nM, 400 nM, 200 nM, 100 nM, 50 nM, and 25 nM). Sample loading time is 100 seconds, and disassociation time is 400 to 800 seconds. Then the biosensor was regenerated in 10 mM glycine HCl (pH1.5) for 15 seconds. A simple one to one Langmuir model was chosen to calculate $k_D$ ($k_{on}$ and $k_{off}$) in Octet Red96 software.

## Co-binding test by Octet

SARS-CoV-S1 protein was immobilized, Ab1 was added with saturated concentration, then change to Ab2 w/ saturated concentration to obtain signal of Ab2 binding to S1-Ab1 complex as Signal 1. For the control group, Ab1 was replaced with buffer, then change to Ab2 with saturated concentration to obtain signal of Ab2 binding to S1 only as Signal 2. Binding rate of Ab2 to S1-Ab1 complex (i.e., binding rate of Ab2 to S1 when co-binding with Ab1) is Signal 1/Signal 2 × 100%.

## Pseudovirus neutralization test

The pseudovirus neutralization assay was done at Institute of System Medicine, Chinese Academy of Science (Suzhou, China). An MLV SARS-CoV-2 S pseudovirus packaged with luciferase encoded vector was mixed with testing antibody, titrating from 15 μg/ml with 5-fold serial dilution, and incubated with cultured HEK293 cells overexpressing ACE2. After 48 hours, the Luciferase Assay Reagent (Promega, USA, E4550) was used as the detection reagent and the relative luminescence unit (RLU) was measured on a SpectraMax plate reader. The $IC_{50}$ was calculated by a 4-parameter fitting in GraphPad Prism.

**Live virus neutralization test.** All mAbs were 2-fold serially diluted in culture medium and was mixed and incubated with SARS-CoV-2 (National Virus Resource) for 1 hour at 37°C. VeroE6 cells (ATCC, CRL-1586, Lot#: 60526234) in 96-well plate were incubated with the mixture containing virus (100 $TCID_{50}$ per well) in presence or absence with diluted antibodies for 1 hour, after which the cells were washed and further incubated in medium for 48 hours. Then, the cells were fixed with 4% paraformaldehyde for 15 minutes and permeabilized with 0.25% Triton X-100 at room temperature, followed by blocking with 5% BSA in PBS at 37°C for 1 hour. Then cells were incubated with an in-house prepared anti-SARS-CoV-2 NP rabbit serum (1:1,000 dilution) as the primary antibody and the Goat Anti-Rabbit IgG H&L (Alexa Fluor 488) (1:500 dilution, Abcam, Cambridge, United Kingdom; ab150077; Lot#: GR3244688-2) as the secondary antibody according to the manufacturer's instruction. Cell nuclei were stained with Hoechst 33258 (Beyotime, Shanghai, China; C1018) for cell counting. Imaging was done on Operetta CLSTM high throughput system (PerkinElmer, Waltham, USA), and number of infected cells and total cells in each well were counted. $IC_{50}$ and $IC_{90}$ were determined by GraphPad Prism 8.0.

## Antibody scFv fragment expression and purification

The single-chain variable fragment (scFv) of PR1077 contains the HC variable domain ($V_H$) and a light chain variable domain ($V_k$) of the antibody, with a $(GGGGS)_3$ peptide linker between $V_H$ and $V_k$. The codon-optimized wild-type cDNA of PR1077 scFv fragment was synthesized, which with a carboxyl-terminal 6×His tag for purification was cloned into pAcGP67 vector with *BamH*I and *Not*I restriction sites using the cloning primers. The sequences of the primers were forward 5′-CATCTACGCCGACGGATCCCAGGTGCAGCTGGTGGAGT-3′

and reverse 5′-GATGATGTGCGGCCGCTCAGTGGTGGTGGTGGTGGT-3′. The antibody scFv fragment was expressed using the Bac-to-Bac baculovirus system. The resulting recombinant expression plasmid was transformed into DH10Bac component cells, and the extracted bacmid was then transfected into Sf9 cells using Cellfectin II Reagent (Invitrogen). The low-titer viruses were harvested and then amplified to generate high-titer virus stock. The viruses were infected Hi5 cells at a density of $2.0 \times 10^6$ cells/ml. The supernatant of cell culture containing the secreted antibody scFv fragment was harvested 60 hours after infection, which was purified preliminarily on a Ni-NTA affinity column (GE Healthcare). The affinity column was washed with 200 mL of wash buffer (25 mM Tris, 200 mM NaCl, 50 mM imidazole, pH 7.0), and the target protein was eluted with elution buffer containing 25 mM Tris, 200 mM NaCl, 1M imidazole, pH 7.0. The protein was further purified on a Superdex 75 16/60 size-exclusion column (GE Healthcare) equilibrated with 25 mM Tris, 200 mM NaCl, pH 7.0. SDS-PAGE analysis revealed over 90% purity of the final purified recombinant protein. Fractions from the single major peak were pooled and concentrated to 10 mg/mL. The scFv of PR953 and PR961 was expressed and purified following the same protocol.

## Crystal screening

The SARS-CoV-2 RBD protein and antibody scFv fragment were mixed at a molar ratio of 1.2:1. The mixture was incubated on ice for 2 hours and further purified by Superdex 75 16/60 size-exclusion column (GE Healthcare). Moreover, 4 mg/mL and 8mg/mL of SARS-CoV-2 RBD/antibody scFv fragments were used for crystal screening by vapor-diffusion sitting-drop method at 16˚C.

The rode-like diffracting crystals of the PR1077-RBD complex appeared after 7 days at the mother liquid containing 1% w/v Tryptone, 0.001 M Sodium azide, 0.05 M HEPES sodium pH 7.0, 12% w/v Polyethylene glycol 3,350. Crystals were cryo-protected in 30% w/v Polyethylene glycol 3,350 and cooled in a dry nitrogen stream at 100 K for X-ray data collection.

The pyramidal diffracting crystals of the PR953-RBD appeared after 7 days at the mother liquid containing 0.8 M Potassium sodium tartrate tetrahydrate, 0.1 M Tris pH 8.5, 0.5% w/v Polyethylene glycol monomethyl ether 5,000. Crystals were cryo-protected in 4 M Sodium formate and cooled in a dry nitrogen stream at 100 K for X-ray data collection.

The Flaky diffracting crystals of the PR961-RBD appeared after 4 days at the mother liquid containing 0.2 M Sodium chloride, 0.1 M Tris sodium pH 8.5, 29% w/v Polyethylene glycol 3,350. Crystals were cryo-protected in 4 M Sodium formate and cooled in a dry nitrogen stream at 100 K for X-ray data collection.

## X-ray data collection, processing, and structure determination

Diffraction data were collected at the Beamline PX06SA of the Swiss Light Source, Paul Scherrer Institute, Villigen, Switzerland (wavelength, 1Å) at 100K for PR1077 and Shanghai Synchrotron Radiation Facility (SSRF) BL17U1 (wavelength, 0.97915 Å) at 100K for PR953 and PR961. Data sets were processed using the MOSFLM [24] or HKL3000 package [26]. Structures were solved by molecular replacement using PHASER [27] with the SARS-CoV-2 RBD structure (PDB ID: 6M0J) [28] and the structures of the Fab fragment available in the PDB with the highest sequence identities. The initial model was built into the modified experimental electron density using COOT [25] and further refined in PHENIX [29]. Model geometry was verified using the program MolProbity. Structural figures were drawn using the program PyMOL [30] (http://www.pymol.org). Epitope and paratope residues, as well as their interactions, were identified by accessing PISA (http://www.ebi.ac.uk/pdbe/prot_int/pistart.html) at the European Bioinformatics Institute.

### In vitro assay to detect antibody-dependent viral entry

In vitro SARS-CoV-2 pseudovirus ADE assays was performed using Raji cells as previously reported [31]. Briefly, $3 \times 10^4$ Raji cells were seeded in each well of 96-well plates coated with 0.01% poly-L-lysine in PBS and cultured for 24 hours. The antibodies were serially diluted 1:2 (maximum concentration, 100μg/ml) in RPMI 1640 for 9 dilutions in total and were incubated with the SARS-CoV-2 pseudovirus for 30 minutes. The mixture was applied onto the Raji cells and cultured for 60 hours. The measurement of luciferase activity was performed as described above using a Firefly Luciferase Assay Kit (Promega). XG005, together with its silenced mutants XG005-GRLR (G236R and L328R in the Fc receptor-binding site), was used as positive and negative control, respectively.

### Viruses

SARS-CoV-2 strain (nCoV-2019BetaCoV/Wuhan/WIV04/2019) was propagated, stored, and titrated as previously described [32,33]. Construction and amplification of AdV-hACE2 were carried out as previously described [34]. All studies on infectious viruses were performed in a Biosafety Level 3 (BSL-3) laboratory.

### Mouse experiments

Eight to 14-week-old IFNAR$^{-/-}$ C57BL/6 mice were inoculated intranasally with $4 \times 10^8$ TCID$_{50}$ of AdV-hCE2. Five days after AdV transduction, mice were inoculated with $1 \times 10^6$ TCID$_{50}$ of SARS-CoV-2 via an intranasal infection route. Weights were monitored daily, and animals were humanely killed at 6 d.p.i., and tissues were harvested.

### qRT-PCR

Tissues were weighted and homogenized with 1 mL TRIzol (Invitrogen), and RNA was extracted following the manufacturer's protocol. qRT-PCR were carried out as described previously [35]. Briefly, cDNA was generated by reverse transcription using PrimerScript RT reagent Kit with gDNA Eraser (Takara, Japan) from 1 μg RNA, and qRT-PCR was performed on the StepOne Plus Real-Time PCR System (Applied Biosystems, USA) with TB Green Premix Ex Taq II (Takara), using the following condition: 95˚C for 5 minutes, followed by 40 cycles at 95˚C for 5 seconds, 54˚C for 30 seconds, and 72˚C for 30 seconds. Primers were designed to target the RBD of spike gene (RBD-F:5′-GCTCCATGGCCTAATATTA-CAAACTTGTGCC-3′, RBD-R:5′-TGCTCTAGACTCAAGTGTCTGTGGATCAC-3′).

### Histochemical staining

The collected lung tissues were fixed in 10% neutral buffered formalin, and the paraffin embedding and sectioning were carried out routinely. formalin-fixed, paraffin-embedded (FFPE) tissue blocks were sectioned (3-μm thick) and stained with HE histochemical.

## Supporting information

**S1 Fig. Selection and characterization of NAbs. (A)** Schematic of in-chip assay on Beacon Optofluidic system to screening for anti-SARS-CoV-2 antibody secreting B cells. Biotinylated antigen (dark blue) was conjugated to a streptavidin-decorated polystyrene bead (dark gray). Single plasma B cells (blue/purple) were loaded into individual NanoPens on the OptoSelect 14K chip in Beacon Optofluidic system, followed by import of a fluorescent secondary antibody-antigen beads mixture. Antibody secreted by plasma B cells binding to antigen was detected by fluorescent anti-rat or anti-mouse IgG secondary antibody (green). **(B)** Bright

 BIOLOGY

SARS-CoV-2 neutralizing antibody structures

field and fluorescent images of NanoPens with single plasma B cells. The plasma B secreting anti-SARS-CoV-2 antibody was identified by increasing fluorescence near the opening of the NanoPen. **(C)** Representative fluorescent images of an FOV on the OptoSelect 14K chip and zoom in view of NanoPens with positive plasma B cells. FOV, field of view; NAb, neutralizing antibody; SARS-CoV-2, Severe Acute Respiratory Syndrome Coronavirus 2.
(TIF)

**S2 Fig. Images of Vero E6 cells infected SARS-CoV-2 treated with antibodies of different concentrations.** Immunofluorescent staining of SARS-CoV-2 NP for in vitro neutralization assays against live SARS-CoV-2 virus in Vero E6 cells. Green (stained SARS-CoV-2 NP) indicates viral infected cells, and blue (Hoechst 33258) represents cell nuclei. SARS-CoV-2, Severe Acute Respiratory Syndrome Coronavirus 2.
(TIF)

**S3 Fig. SARS-CoV-2 live virus neutralization activity test of isolated mAbs.** The mixtures of SARS-CoV-2 and serially diluted antibodies were added to Vero E6 cells. After 4 days incubation, $IC_{50}$ were calculated by fitting the CPE from serially diluted antibody to a sigmoidal dose- response curve. The $IC_{50}$ were labeled accordingly. **(A)** SARS-CoV-2 live virus neutralization activity test of 11 mAbs from H2L2 mice. **(B)** SARS-CoV-2 live virus neutralization activity test of 14 mAbs from BALB/c mice. All the data of this figure can be found in the S11 and S12 Data files. mAb, monoclonal antibody; SARS-CoV-2, Severe Acute Respiratory Syndrome Coronavirus 2.
(TIF)

**S4 Fig. Epitope mapping by competitive ELISA.** Coated antibody was bound to the substrate in a 96-well high binding plate. Competitor antibody and biotinylated SARS-CoV-2 S1 protein were mixed and incubated for 30 minutes before adding to the plated. After 1-hour incubation, an Alexa 488 streptavidin was used to detect the biotinylated SARS-CoV-2 S1 protein bound to coated antibody. The percentage of inhibition was calculated. Blue indicates no competition, and red indicates competition. The groups 1–4 at the bottom indicates clusters of antibody epitopes defined by the competitive ELISA results. All the data of this figure can be found in the S13 Data file. ELISA, enzyme-linked immunosorbent assay; SARS-CoV-2, Severe Acute Respiratory Syndrome Coronavirus 2.
(TIF)

**S5 Fig. Characterization of PR1077 as potential therapeutic candidate. (A)** Side or top view of PR1077 binding epitope (colored red) on SARS-CoV-2 spike trimer in "all open form" (superimposed to the spike trimer structure, PDB code: 7CAK) or "all RBD closed state" (superimposed to the spike trimer structure, PDB code: 6ZP0). **(B)** ELISA binding curves of 22 RBD mutants. PR1077 binds strongly to a wide range of RBD mutants. All the data of this figure can be found in the S14 Data file. **(C)** PR1077 can effectively neutralize a range of SARS-CoV-2 mutant pseudovirus. All the data of this figure can be found in the S15 Data file. ELISA, enzyme-linked immunosorbent assay; RBD, receptor-binding domain; SARS-CoV-2, Severe Acute Respiratory Syndrome Coronavirus 2.
(TIF)

**S6 Fig. Potential ADE test of PR1077, XG005 (positive control), and XG005-GRLR (negative control).** Raji cells as a FcγRII-bearing human B lymphoblast cell line, to study the antibody-dependent viral entry. The antibody-dependent entry of SARS-CoV-2 pseudovirus in the presence of PR1077, XG005, and XG005-GRLR was measured at various antibody concentrations. All the data of this figure can be found in the S16 Data file. ADE, antibody-dependent

enhancement; SARS-CoV-2, Severe Acute Respiratory Syndrome Coronavirus 2.
(TIF)

**S7 Fig. The epitopes of hACE2 and selected NAbs are shown in the surface representation.** The SARS-CoV-2 RBD is colored in gray and displayed in surface representation. The binding epitope for hACE2 (PDB code: 6M0J), PR1077 (PDB code:7DEO), PR953 (PDB code:7DEU), PR961 (PDB code:7DET), P4A1 (PDB code:7CJF), CB6 (PDB code: 7C01), BD-629 and BD-368 (PDB code: 7CHC), REGN10933, and REGN10987(PDB code: 6XDG) are displayed in different colors. hACE2, human angiotensin-converting enzyme 2; NAb, neutralizing antibody; RBD, receptor-binding domain; SARS-CoV-2, Severe Acute Respiratory Syndrome Coronavirus 2.
(TIF)

**S8 Fig. Sequence alignment of PR1077 to other reported antibodies from the IGHV3-30 or 3–33 germline. (A)** Alignment of the HC variable domain sequence of PR1077 with C002, C135, EY6A, REGE10987 and the germline IGHV3-30, IGHV3-33 sequence. **(B)** Alignment of the light chain variable domain sequence of PR1077 with the germline IGKV2-28 sequence. **(C)** Germline usage comparison of the HC and light chain discussed in this paper. HC, heavy chain.
(TIF)

**S9 Fig. Alignment of selected NAbs derived from VH3-33 and VH3-30 germline.** PR1077 (HC: slate blue, light chain: red), C102 (HC: dark blue, light chain: golden yellow), EY6A (yellow orange), C135 (cyan), and REGN10987 (limon green) are superimposed. SARS-CoV-2 spike RBD are shown in surface and colored gray. NAb, neutralizing antibody; RBD, receptor-binding domain.
(TIF)

**S1 Table. Summary of NAbs binding, blocking, and neutralization assays.** The binding kinetics of these mAbs were further assessed by BLI assays, and equilibrium constant ($K_D$) values were calculated. BLI, bio-layer interferometry; mAb, monoclonal antibody; NAb, neutralizing antibody.
(XLSX)

**S2 Table. Data collection and refinement statistics.** *Numbers in the brackets are for the highest resolution shell. $R_{merge} = \Sigma_h\Sigma_l|I_{ih}-<I_h>|/\Sigma_h\Sigma_I <I_h>$, where $<I_h>$ is the mean of the observations $I_{ih}$ of reflection h. $R_{work} = \Sigma (||F_p (obs)|-|F_p (calc)||)/\Sigma|F_p (obs)|$; $R_{free}$ is an R factor for a preselected subset (5%) of reflections that was not included in refinement.
(XLSX)

**S3 Table. Residues contributed to interaction between NAbs/SARS-CoV-2-RBD. (a)** The H and S in parentheses refer to hydrogen bonds and salt bridges, respectively. **(b)** Light chain residues are listed in italics in square bracelets. NAb, neutralizing antibody; RBD, receptor-binding domain; SARS-CoV-2, Severe Acute Respiratory Syndrome Coronavirus 2.
(XLSX)

**S1 Data. The individual numerical values for the following figure panel: Fig 1B.**
(XLSX)

**S2 Data. The individual numerical values for the following figure panel: Fig 1C.**
(XLSX)

**S3 Data. The individual numerical values for the following figure panel: Fig 2A.**
(XLSX)

**S4 Data. The individual numerical values for the following figure panel: Fig 2B.**
(XLSX)

**S5 Data. The individual numerical values for the following figure panel: Fig 2C.**
(XLSX)

**S6 Data. The individual numerical values for the following figure panel: Fig 2D.**
(XLSX)

**S7 Data. The individual numerical values for the following figure panel: Fig 2E.**
(XLSX)

**S8 Data. The individual numerical values for the following figure panel: Fig 2F.**
(XLSX)

**S9 Data. The individual numerical values for the following figure panel: Fig 5.**
(XLSX)

**S10 Data. The individual numerical values for the following figure panel: Fig 5C.**
(XLSX)

**S11 Data. The individual numerical values for the following figure panel: S3A Fig.**
(XLSX)

**S12 Data. The individual numerical values for the following figure panel: S3B Fig.**
(XLSX)

**S13 Data. The individual numerical values for the following figure panel: S4 Fig.**
(XLSX)

**S14 Data. The individual numerical values for the following figure panel: S5B Fig.**
(XLSX)

**S15 Data. The individual numerical values for the following figure panel: S5C Fig.**
(XLSX)

**S16 Data. The individual numerical values for the following figure panel: S6 Fig.**
(XLSX)

## Acknowledgments

We thank the running team of Biosafety Mega-Science, Wuhan, Chinese Academy of Sciences and the staff of BL-17U1, Shanghai Synchrotron Radiation Facility.

## Author Contributions

**Conceptualization:** Zhiyong Lou, Hongkai Zhang, Xinwen Chen, Louis Liu, Zihe Rao, Yu Guo.

**Data curation:** Hengrui Hu, Jun Wu.

**Formal analysis:** Hengrui Hu, Yu Guo.

**Funding acquisition:** Hongkai Zhang, Louis Liu, Zihe Rao, Yu Guo.

**Investigation:** Dan Fu, Guangshun Zhang, Yuhui Wang, Zheng Zhang, Hengrui Hu, Shu Shen, Jun Wu, Bo Li, Xin Li, Yaohui Fang, Jia Liu, Qiao Wang, Yunjiao Zhou, Wei Wang, Zhonghua Lu, Xiaoxiao Wang, Cui Nie, Yujie Tian, Yuan Wang, Xingdong Zhou, Feng Yu, Chen Zhang, Changjing Deng, Liang Zhou, Guangkuo Guan, Na Shao, Fei Deng, Xinwen Chen, Manli Wang, Louis Liu.

**Methodology:** Dan Fu, Guangshun Zhang, Yuhui Wang, Zheng Zhang, Hengrui Hu, Shu Shen, Jun Wu, Bo Li, Xin Li, Yaohui Fang, Jia Liu, Qiao Wang, Yunjiao Zhou, Wei Wang, Yufeng Li, Zhonghua Lu, Xiaoxiao Wang, Cui Nie, Yujie Tian, Da Chen, Yuan Wang, Xingdong Zhou, Feng Yu, Chen Zhang, Changjing Deng, Liang Zhou, Guangkuo Guan, Na Shao, Fei Deng, Manli Wang, Louis Liu.

**Project administration:** Yufeng Li, Da Chen, Zhiyong Lou, Fei Deng, Zihe Rao, Yu Guo.

**Resources:** Qiao Wang, Zhiyong Lou, Xinwen Chen.

**Software:** Guangshun Zhang, Jun Wu, Xin Li, Zhonghua Lu, Da Chen, Qisheng Wang, Feng Yu.

**Supervision:** Qiao Wang, Xiaoxiao Wang, Yuan Wang, Qisheng Wang, Fei Deng, Hongkai Zhang, Manli Wang, Zihe Rao, Yu Guo.

**Validation:** Dan Fu, Guangshun Zhang, Qisheng Wang.

**Visualization:** Guangshun Zhang, Jun Wu, Xin Li, Da Chen, Qisheng Wang.

**Writing – original draft:** Jun Wu, Hongkai Zhang, Xinwen Chen, Zihe Rao, Yu Guo.

**Writing – review & editing:** Manli Wang, Yu Guo.

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
