## [Editor Report · Decision Letter 0]

13 Jan 2021

Dear Dr. Guo, 

Thank you for submitting your manuscript entitled "Structural basis for SARS-CoV-2 neutralizing antibodies with novel binding epitopes" for consideration as a Research Article by PLOS Biology.

Your manuscript has now been evaluated by the PLOS Biology editorial staff [as well as by an academic editor with relevant expertise] and I am writing to let you know that we would like to send your submission out for external peer review.

Please re-submit your manuscript within two working days, i.e. by Jan 15 2021 11:59PM.

Kind regards,

Paula

---

Associate Editor

PLOS Biology

---

## [Decision Letter · Decision Letter 1]

23 Feb 2021

Dear Dr. Guo,

Thank you very much for submitting your manuscript "Structural basis for SARS-CoV-2 neutralizing antibodies with novel binding epitopes" for consideration as a Research Article at PLOS Biology. Your manuscript has been evaluated by the PLOS Biology editors, an Academic Editor with relevant expertise, and by several independent reviewers.

In light of the reviews (below), we are pleased to offer you the opportunity to address the comments from the reviewers in a revised version that we anticipate should not take you very long. We will then assess your revised manuscript and your response to the reviewers' comments and we may consult the reviewers again.

In particular, reviewer #1 says that you don’t provide sufficient experimental data to support the conclusion that the antibody binds to the RBD with “open” or “all close state”, and wants you to validate the immunofluorescence results using a published antibody. Please, also address the rest of the reviewers' concerns. 

We expect to receive your revised manuscript within 1 month.

**IMPORTANT - SUBMITTING YOUR REVISION**

*Resubmission Checklist*

*Published Peer Review*

*PLOS Data Policy*

*Blot and Gel Data Policy*

Sincerely,

Paula

---

Associate Editor,

pjaureguionieva@plos.org,

PLOS Biology

REVIEWS:

Reviewer #1: Therapeutic antibodies.

Reviewer #2: Structure-based vaccine design and host immune response-pathogens interactions.

Reviewer #1: Major Points

1. The authors suggested monoclonal antibody PR1077 binds to the RBD of spike protein in either "open" or "all close state". The manuscript did not provide sufficient experimental data to support this conclusion. As shown in Fig S7, the binding epitope of PR1077 largely overlaps with hACE2 binding region. A CryEM structure of SARS-CoV-2 spike protein trimer has been recently published (Science 2020 367(6483):1260-1263). The authors should specify the critical binding epitopes of PR1077 exposed for surface binding in both "open" or "down" conformations (all close state).

2. Immunofluorescence staining assay was used instead of the classic PRNT assay. In this case, a reliable primer antibody for staining is critical to validate the assay. An in-house prepared and uncharacterized anti-SARS-CoV-2 rabbit serum sample was used to visualize virus infected cells. The authors should use other published neutralizing antibodies to confirm their results (Fig 2F).

3. Has PR1077 been tested against any new SARS-COV-2 variants? This would significantly increase the impact of this story.

Minor points:

1. Several COVID-19 vaccines have been already approved. The statement "Currently, there is no approved vaccines or therapeutics against COVID-19" in the abstract needs to be revised.

2. A statistical test needs to be added in Fig 5B and C for comparison between the PR1077 treated group and PBS control.

3. More discussion of PR961 neutralization mechanism could be added. PR961 appeared to be a non-receptor blocking neutralizing antibody, which is unique as compared to other published neutralizing antibodies.

Reviewer #2: Fu et al. reported the use of a transgenic mouse platform to generate antibodies against the RBD of the SARS-CoV-2 spike protein. The authors identified 25 potently neutralizing RBD-specific antibodies and determined the structure of three antibodies with novel epitopes on the RBD. They further showed the prophylactic or therapeutic efficacy of the PR1077 antibody against SARS-CoV-2 infection. In addition, the binding and neutralization ability of PR1077 against numerous emerging RBD mutants was demonstrated. I recommend acceptance with minor revisions. 

Minor revisions:

1) In paragraph 3 of the "Selection and characterization of the purified antibodies" section, the authors claimed that the VH3-33 germline gene usage is among the most frequently used but do not include a citation. Recent reviews do not show this germline gene among the most used but do show the similar VH3-30 gene among the top 5-6. The similarity of the germline genes was later mentioned in the Discussion section when the comparison to published VH3-30 antibody structures was discussed. 

2) Figure 1D is never cited in the manuscript.

---

## [Decision Letter · Decision Letter 2]

11 Mar 2021

Dear Dr. Guo,

Thank you for submitting your revised Research Article entitled "Structural basis for SARS-CoV-2 neutralizing antibodies with novel binding epitopes" for publication in PLOS Biology. I have now obtained advice from reviewer #1 and have discussed their comments with the Academic Editor. 

Based on the reviews, we will probably accept this manuscript for publication, provided you satisfactorily address the following data and other policy-related requests.

DATA POLICY:

Regardless of the method selected, please ensure that you provide the individual numerical values that underlie the summary data displayed in the following figure panels as they are essential for readers to assess your analysis and to reproduce it: Figure 1B, 1C, 2A, 2B, 2C, 2D, 2E, 2F, 5B, 5C, S3A, S3B, S4, S5B, S5C, and S6.

Please also provide size bars for the microscopy pictures in figure S2.

**Please also ensure that figure legends in your manuscript include information on where the underlying data can be found, and ensure your supplemental data file/s has a legend.**

We expect to receive your revised manuscript within two weeks.

*Published Peer Review History*

*Early Version*

Sincerely,

Paula

---

Associate Editor,

pjaureguionieva@plos.org,

PLOS Biology

Reviewer remarks:

Reviewer #1: The revised version addressed reviewer's concerns. It is recommended for acceptance.

---

## [Editor Report · Decision Letter 3]

26 Mar 2021

Dear Dr. Guo,

On behalf of my colleagues and the Academic Editor, Ken Cadwell, I am pleased to say that we can in principle offer to publish your Research Article "Structural basis for SARS-CoV-2 neutralizing antibodies with novel binding epitopes" in PLOS Biology, provided you address any remaining formatting and reporting issues.

Please ensure that figure legends in your manuscript include information on where the underlying data can be found. 

The rest of the formatting and reporting issues will be detailed in an email that will follow this letter and that you will usually receive within 2-3 business days, during which time no action is required from you. Please note that we will not be able to formally accept your manuscript and schedule it for publication until you have made the required changes.

PRESS

We frequently collaborate with press offices. If your institution or institutions have a press office and is planning to promote your findings, we would be grateful if they could coordinate with biologypress@plos.org. If you have not yet opted out of the early version process, we ask that you notify us immediately of any press plans so that we may do so on your behalf.

Thank you again for supporting Open Access publishing. We look forward to publishing your paper in PLOS Biology. 

Sincerely, 

Paula

---

Paula Jauregui, PhD 

Associate Editor 

PLOS Biology